# Mosquitoes and Mosquito-Borne Diseases in Vietnam

**DOI:** 10.3390/insects13121076

**Published:** 2022-11-22

**Authors:** Ly Na Huynh, Long Bien Tran, Hong Sang Nguyen, Van Hoang Ho, Philippe Parola, Xuan Quang Nguyen

**Affiliations:** 1Aix Marseille Univ, IRD, AP-HM, SSA, VITROME, 13005 Marseille, France; 2IHU-Méditerranée Infection, 19-21 Boulevard Jean Moulin, 13005 Marseille, France; 3Institute of Malariology, Parasitology and Entomology, Quy Nhon (IMPE-QN), MoH Vietnam, Zone 8, Nhon Phu Ward, Quy Nhon City 590000, Vietnam

**Keywords:** *Aedes*, *Anopheles*, *Culex*, Anophelinae, Culicinae, Culicidae, vector, Vietnam

## Abstract

**Simple Summary:**

Vietnam is one of the tropical countries of Asia where individuals are at high risk of attaining mosquito-borne diseases. Due to rapid urbanization in Vietnam, most of the major cities have immense population growth, along with inadequate control measures over mosquitoes. These factors contributed to a sudden increase in the population of disease vectors that lead to cyclical epidemics of mosquito-borne diseases. This review paper aims to (i) provide a complete checklist of Vietnamese mosquitoes, (ii) provide an overview of mosquito-borne diseases in Vietnam, and (iii) preventive measures for mosquitoes in Vietnam. We list 281 mosquito species, belonging to 42 subgenera of 22 genera. We found that three genera, namely, *Anopheles*, *Aedes*, and *Culex* are found to be potential vectors for mosquito-borne diseases in Vietnam. We found dengue and malaria are the most common mosquito-borne diseases in Vietnam with about 320,702 cases and 54 deaths in the 2019 outbreak and 4548 clinical cases and six deaths, respectively. We suggest that mosquito-borne diseases could be effectively controlled and prevented through mechanical, chemical, biological, and genetic methods.

**Abstract:**

Mosquito-borne diseases pose a significant threat to humans in almost every part of the world. Key factors such as global warming, climatic conditions, rapid urbanisation, frequent human relocation, and widespread deforestation significantly increase the number of mosquitoes and mosquito-borne diseases in Vietnam, and elsewhere around the world. In southeast Asia, and notably in Vietnam, national mosquito control programmes contribute to reducing the risk of mosquito-borne disease transmission, however, malaria and dengue remain a threat to public health. The aim of our review is to provide a complete checklist of all Vietnamese mosquitoes that have been recognised, as well as an overview of mosquito-borne diseases in Vietnam. A total of 281 mosquito species of 42 subgenera and 22 genera exist in Vietnam. Of those, *Anopheles*, *Aedes*, and *Culex* are found to be potential vectors for mosquito-borne diseases. Major mosquito-borne diseases in high-incidence areas of Vietnam include malaria, dengue, and Japanese encephalitis. This review may be useful to entomological researchers for future surveys of Vietnamese mosquitoes and to decision-makers responsible for vector control tactics.

## 1. Introduction

Mosquitoes are vectors of disease and can transmit many infectious pathogens (arboviruses, filariae, and protozoans) that cause common and emerging diseases, specifically malaria, dengue, Zika, chikungunya, and Japanese encephalitis, between humans or from other creatures to humans [1]. Mosquito-borne diseases are widely distributed in almost all countries of the world, but are mainly concentrated in tropical and subtropical regions of the globe with warm and humid climates [2]. There are 3563 valid species, belonging to three genera (*Anopheles*, *Aedes*, and *Culex*) within two subfamilies (Anophelinae and Culicinae) [3,4]. The mosquito vectors of the *Anopheles* genus can transmit malaria and filariasis, while the *Aedes* genus can transmit dengue virus (DENV), chikungunya virus (CHIKV), Zika virus (ZIKV), yellow fever virus (YFV), and the *Culex* genus can transmit filariasis, Japanese encephalitis virus (JEV), West Nile virus (WNV), and Rift Valley Fever (RVF) [5,6]. According to the World Health Organization (WHO, 2020), every year about 219 million malaria cases and 96 million symptomatic dengue cases, leading to more than 445,000 and 40,000 deaths, respectively, are recorded globally [4,7]. There are no potential vaccines available for most mosquito-borne diseases. The WHO has, therefore, declared that the key preventive measures for controlling mosquito-borne diseases are a change of behaviour, public awareness, and environmental management [7].

Vietnam is a tropical country in Asia where individuals are at high risk of acquiring mosquito-borne diseases such as malaria, dengue, Zika, Japanese encephalitis, and lymphatic filariasis [8]. Although malaria was previously endemic in many parts of Vietnam, it has recently been significantly reduced following vector control efforts by the Vietnamese government. These measures included prompt case detection and treatment with pyronaridine-artesunate (Pyramax) which is used for diversifying anti-malarial therapy in artemisinin- and piperaquine-resistant *Plasmodium falciparum* parasite regions [9], the widespread distribution of insecticide-treated bed nets (ITNs), long-lasting insecticide-treated bed nets/long-lasting insecticide-treated hammock nets (LLTNs/LLHNs), and the regular the use of pyrethroids through indoor residual spraying (IRS) in households [10,11,12,13,14,15]. However, malaria continues to persist as an important public health threat in the Central Highlands and the southern forested and mountainous regions [16,17], where almost all ethnic minority groups live in impoverished socio-economic conditions. It mainly affects people such as farmers and/or forest-goers, as well as temporary migrants moving from endemic regions [18]. In addition, vector resistance to chemical insecticide, has been found in every main species. *Anopheles minimus* sensu lato (s.l.) was found to be pyrethroid-resistant in northern Vietnam, *Anopheles dirus* s.l. showed possible tolerance to type II pyrethroids in central Vietnam, and *Anopheles epiroticus* showed high resistance to pyrethroids in the Mekong Delta region [19,20]. The drug-resistant malaria strains in particular pose a major potential threat to extensive malaria control and elimination strategies planned for the whole country before 2030 [13,21,22]. Cases of the multidrug resistance of the *P*. *falciparum* malaria parasite in Vietnam and other southeast Asian countries such as Laos, Cambodia, Thailand, and Myanmar, have been published [16,23,24]. Dengue fever (DF) is the fastest spreading mosquito-borne arboviral infection in the whole world, which is transmitted to humans through the bites of infected female *Aedes aegypti* or *Aedes albopictus* mosquitoes [25]. Vietnam has been ranked among the five countries with the heaviest dengue burden in the Asian-Pacific region [26,27]. Due to rapid urbanisation, most of the major cities in Vietnam have experienced immense population growth, coupled with inadequate mosquito control measures. Additionally, the country faces the risk of climate change [28,29] and extreme weather variability, with widespread flooding due to rising sea levels. Other unexpected outcomes of climate change include rising temperatures and decreasing rainfall. These factors have contributed to a sudden increase in the population of disease vectors that lead to cyclical epidemics of mosquito-borne diseases [30].

This is the first review that provides a complete checklist of Vietnamese mosquitoes along with a comprehensive and systematic overview of mosquito-borne diseases in Vietnam. We hope it may be useful to decision-makers responsible for vector control strategies and to researchers for future surveys on mosquitoes.

## 2. Mosquitoes and Medical Importance in Vietnam

In Vietnam, detailed studies on mosquitoes (particularly on the *Anopheles*, *Culex*, and *Aedes* species) were initiated after the first outbreak of the dengue virus in 1959 [30]. However, the first detailed report on mosquitoes and their relationship to human diseases in Vietnam was proposed by Parrish in 1968 [31]. In this report, Parrish identified 94 different mosquito species from ten United States Air Force (USAF) installations in Vietnam between June 1966 and June 1968. Of these, 22 species were reported as main vectors of human diseases [31]. In 1970, Reinert [32] and Tyson [33] improved the catalogue of *Culex* and *Aedes* mosquitoes in SEA countries including Vietnam, Malaysia, Indonesia, and the Philippines.

In 1974, Thoa [34] collected and examined around 107,000 mosquitoes in and around poultry houses and piggeries in Saigon city (now known as Ho Chi Minh City) between 1972–1973 and noted that the *Culex* species (particularly *Culex tritaeniorhynchus*) was predominant. Later, Thuan [35] studied and reported the evolution of *Anopheles* mosquitoes in the Da Nang, Quang Nam, Phu Khanh, Nghia Binh, and Thuan Hai provinces, between 1975 and 1986. Since then, multiple studies on the diversity of medical mosquito vectors in Vietnam have been published [36]. Bui et al. (2008) listed 191 species and subspecies, 28 subgenera and 35 genera, and the authors compared their research with those published from US military sources during the Vietnam war. However, these data were not exhaustive, as they did not update all the reported research by domestic authors [37]. More recently, Nguyen (2015) listed 255 species in 21 genera and 42 subgenera. The author’s list was based on research papers published by domestic and foreign researchers [38]. Nevertheless, none of them are documented as a literature review of existing mosquito species, along with their associated mosquito-borne diseases. Hence, in this review, we provide a complete checklist of all recognised mosquitoes in Vietnam. It was compiled from multiple reliable studies published both nationally and internationally by Parrish (1969) and Grothaus et al. (1971) [31,32], Reinert (1973) [39], Vu (1984) [36], Harbach (2007) [40], Bui et al. (2008) [37] and Nguyen (2015) [38].

Our research reported the existence of a total of 281 Vietnamese mosquito species in 42 subgenera and 22 genera, of which the main vectors were 66 *Anopheles*, 51 *Aedes*, 50 *Culex*, one *Aedeomyia*, 12 *Armigeres*, three *Coquillettidia*, one *Ficalbia*, seven *Heizmannia*, one *Hodgesia*, one *Kimia*, four *Lutzia*, three *Malaya*, nine *Mansonia*, three *Mimomyia*, 24 *Ochlerotatus*, three *Orthopodomyia*, three *Topomyia*, four *Toxorhynchites*, six *Tripteroides*, one *Udaya*, 20 *Uranotaenia*, and eight *Verrallina* species [31,36,37,38,39,40]. These findings are presented in Table 1 and Table 2. The distribution of the primary mosquito vectors in Vietnam and breeding and resting sites are shown in Figure 1 and Figure 2.

## 3. Mosquito-Associated Microorganims and Human Diseases

### 3.1. Malaria

Malaria is a protozoal infection of erythrocytes and ranks fourth among parasitic diseases affecting birds, reptiles, humans, and other mammals [4,5] Human malaria is also known as ague, marsh fever, paludism, and intermittent fever [4]. Of the 172 known *Plasmodium* species, only four, namely *P*. *falciparum*, *Plasmodium vivax*, *Plasmodium malariae*, and *Plasmodium ovale*, are commonly known to infect humans with malaria [5]. *P*. *falciparum* has caused the greatest morbidity and mortality to date, with several hundreds of millions of clinical malaria cases [41].

In Vietnam, malaria became a serious public health problem in the late 1980s and early 1990s [42]. In 1991, the National Malaria Control Programme (NMCP), based on ITNs, IRS, and early diagnosis and treatment of malaria (EDTM), was approved. Since then, there has been a reduction in malaria cases between 1991, when there were estimated to be 1,674,000 uncomplicated and severe clinical cases and 4650 deaths, and 2017, when there were estimated to be 4548 clinical cases and six deaths [42,43,44]. Nevertheless, as is the case in many countries in the Asia-Pacific region, Vietnam faces the challenges of rapidly evolving multi-drug resistant parasites [45], changes in the outdoor feeding and early biting and resting behaviour of some mosquitoes, as well as insecticide-resistant mosquitoes, which impact the malarial control and elimination strategies which were expected to eradicate the disease by 2030.

Early in the 19th century, Leger (1910) published 15 species of *Anopheles* mosquitoes in Vietnam. Gallilard and Dang (1946) built the taxonomic key of 22 *Anopheles* mosquito species. Tran (1995), published a list of 55 species nationwide [46]. To date, 64 *Anopheles* species have been identified, but, only 15 *Anopheles* species have formally been recognised as likely to transmit malaria [47].

Three vectors have been incriminated as playing the role of primary vector in malaria transmission, namely *An. dirus* Peyton and Harrison, *An. minimus* Theobald, and *An. epiroticus* Linton and Harbach (Sundaicus complex) [19,48,49,50]. Other secondary vectors are *Anopheles aconitus* Dönitz, *Anopheles campestris* Linnaeus, *Anopheles culicifacies* Giles, *Anopheles indefinitus* Ludlow, *Anopheles interruptus* Puri, *Anopheles jeyporiensis* James, *Anopheles maculatus* Theobald, *Anopheles lesteri* Baisas and Hu, *Anopheles nimpe* Nguyen, Tran and Harbach, *Anopheles sinensis* Wiedemann, *Anopheles subpictus* Grassi, and *Anopheles vagus* Dönitz, which may be considered to contribute towards the transmission of malaria outside of forested regions [11].

*Anopheles dirus* sensu stricto (s.s.) Peyton and Harrison (Figure 3a), 1979, spread to forested and mountain areas from the central and south-eastern regions of Vietnam [51]. *An. dirus* s.s. belongs to the *Anopheles* (*Cellia*) *leucosphyrus* group, with 20 member species [52]. This species, previously known as *Anopheles balabacensis*, comes from Vietnam and other SEA countries [44]. Recently, it was formally named *An. dirus* s.s. (species A) [52]. Of the four *Plasmodium* species reported in Vietnam and known to be true parasites of humans, *P*. *falciparum* (64%) and *P*. *vivax* (35%) [53] have commonly been found, while *P*. *ovale*, *P*. *malariae* are also present but at low levels [53,54,55]. In 2009, a fifth parasite species was discovered, *Plasmodium knowlesi*, a macaque parasite that was recognised for the first time in a case of human malaria in a nine-year-old child in Khanh Phu in the Khanh Hoa province of Vietnam [56]. Interestingly, *An. dirus* was incriminated as the vector for carrying sporozoites of *P. knowlesi*, *P. falciparum*, and *P. vivax*, in a paper published by Charmand [57]. *An. minimus* s.l. (Figure 3b) belongs to the Minimus subgroup. The *An. minimus* complex comprises three species, of which two are known to occur in the Greater Mekong Subregion (GMS), namely *An. minimus* Theobald (formerly *An. minimus* A), and *Anopheles harrisoni* Harbach and Manguin (formerly *An. minimus* C) [19]. *An. minimus* s.l. is widespread in hilly and forested areas nationwide. These mosquitoes breed in small, slow-flowing streams with aquatic vegetation, and clear water in full sunlight [58]. *An. epiroticus* (formerly *Anopheles sundaicus* species A) (Figure 3c), which is distinguished by brackish water populations in sunlight, only occurs in coastal areas of southern Vietnam [59].

### 3.2. Dengue

Dengue or dengue fever is the most prominent mosquito-borne virus, leading to approximately 100 million symptomatic cases in about 100 tropical-zone countries every year [60,61]. According to the WHO (2021), the number of dengue cases increased more than eight-fold from 505,430 cases in 2000 to 5.2 million cases in 2019, with an increase in the death rate from 960 in 2000 to 4032 in 2015 [60]. It is estimated that approximately 70% of these cases were in Asia, with the most affected countries being, in order, the Philippines (420,000), Vietnam (320,000), Malaysia (131,000), and Bangladesh (101,000) [60]. The dengue virus (family *Flaviviridae*) has five distinct serotypes (DENV-1 to DENV-5) and serotypes 1–4 are widely spread globally [62]. Infection with any of these five serotypes can cause illnesses such as dengue fever and dengue shock syndrome [60]. Generally, the dengue virus is transmitted in humans through bites of the female *Aedes* mosquito.

Vietnam is the country that is second most affected by the dengue virus in Asia, as well as in the world, and the first dengue case was reported in 1959 [8,60,63]. However, the largest dengue outbreak was reported in 2019, with about 320,702 cases and 54 deaths between January and December [63]. According to a report from the Vietnamese Ministry of Health, a 2.5-fold increase in the number of dengue cases was noted in the 2019 dengue outbreak, compared to cases in 2018 [63]. It has been reported that the dengue virus (mostly serotypes 1–4) has been highly distributed across Vietnam, from the southern region into the central and northern regions of the country. For instance, Nha Trang city in the central region of Vietnam recorded 12,655 dengue fever cases between 2006 and 2016 [64]. The capital city, Hanoi, (northern region) has witnessed many significant dengue outbreaks over the last decade. The largest outbreak was recorded in 2017 with 37,651 cases and seven deaths [8]. The incidence rate of the dengue virus varies for each province and region, and the most affected cases are recorded in the southern part of the country compared to other regions [65,66]. To date, Vietnam has recorded an increase of about 15,000 dengue cases compared to statistics in the past 10 days, raising the total number of cases in 2022 as of July 2022 to 92,000 cases and 36 deaths [67,68]. The main hotspot areas for dengue fever are in southern Vietnam, specifically Ho Chi Minh with 21,750 cases, an increase of more than 181% compared to the same period last year, and leading to nine deaths [67,69], followed by the provinces of Binh Duong (5000 cases and eight deaths) [70], Dong Nai (3500 cases and three deaths), and An Giang (4400 cases) [71]. These provinces have all witnessed an increasing number of infections. Due to the high population growth and rapid urbanisation, the biggest city, Ho Chi Minh, is well-known as a critical urban centre for the transmission of the dengue virus.

*Aedes aegypti* (Figure 4a) and *Ae. albopictus* (Figure 4b) are two main vectors that cause dengue fever in Vietnam. *Ae. aegypti* is a dominant vector in urban regions of Vietnam, mainly breeding in flower vases, jars, and plastic buckets. *Ae. albopictus* breeds with lower density in both urban and peri-urban areas than *Ae. aegypti*, due to climate change, migration, and urbanisation. The factors influencing the distribution of the dengue virus in Vietnam have been found to be industrialisation, urbanisation, and climate change [72].

### 3.3. Zika Virus Infection

Zika virus is an arbovirus (*Flavivirus* genus) with a rapid geographic spread [73]. It was first isolated from the blood samples of a rhesus monkey, in the Ziika Forest of Uganda, in 1947, during epidemiological studies of yellow fever [74,75]. However, the first case in humans was isolated in a Nigerian female (10 years old) in 1954 [76] and, outside the African continent, it was first identified in SEA (Malaysia 1966) and Indonesia (1977) [77,78]. However, the largest outbreak in humans was reported on the island of Yap in 2007, where 75% of the population was infected [79,80]. Later, new epidemics broke out in French Polynesia (in late 2013) [81], the South Pacific (during 2014 and 2016), the region of the Americas (in 2015), and Brazil (in 2015), leading the WHO to declare “congenital Zika syndrome” as a global public health emergency in February 2016 [78].

Serological screening during these epidemics identified that the *Aedes* species, including *Aedes hensilli* in Yap island, *Aedes polynesiensis* and *Ae. aegypti* in French Polynesia, and *Aedes albopictus* and *Ae. aegypti* in much of the Americas, are the probable vectors of Zika infection [73]. However, *Ae. aegypti* is the leading vector in urban areas, while *Ae. albopictus* occurs in both urban and rural areas [82]. In addition, the Zika virus has also been rarely isolated from *Anopheles*, *Culex*, and *Mansonia* species [83]. In recent ages, some evidence has been found that the Zika virus can also be transmitted by perinatal transmission, transfusions, and sexually [75]. The general clinical symptoms of the Zika virus are arthritis or arthralgia, conjunctivitis, fever, Guillain-Barré syndrome, headache, muscle and joint pain, myalgia, rash, retro-orbital pain, tiredness, and vomiting [84,85].

Vietnam is one of the top five countries that have been recognised as sentinel indicators for Zika virus transmission [86]. In Vietnam, the first Zika infection was identified by Real-Time PCR in two children from the Long An province and Ho Chi Minh City in 2013 [87]. Later, it was identified in two females from Nha Trang city and Ho Chi Minh City in April 2016 [88]. During this first outbreak in 2016, around ten provinces in the Central Highlands and southern regions were strongly affected with 212 cases [88]. Furthermore, the number of cases dropped significantly to 24 cases in 2017, and one case in 2018 [88,89]. All cases reported in Vietnam were recognised to be local, vector-borne infections [89]

In Vietnam, the *Aedes* species is the main vector reported to cause transmission of the Zika virus [89]. The entomological report during the 2016 outbreak noted that the Zika virus was mainly transmitted by the *Ae. aegypti* vector, which mostly survives in water stored at home [89,90]. In addition, the significant growth of the density of the vector was found to be due to (a) the impact of urbanisation; (b) climate change; (c) lack of awareness of the importance of eliminating the breeding areas of mosquitoes [91].

### 3.4. Japanese Encephalitis

Japanese encephalitis is caused by a *Flavivirus*, which is the most common vector-borne disease in Asia and Pacific countries. It affects 67,900 people every year all over the globe [91]. Around 75% of these cases are reported among children 0–14 years old. The general clinical symptoms of viral encephalitis are disorientation, headache, high fever, neck stiffness, seizures, spastic paralysis, and coma [91,92]. Some serological screening studies have reported that the encephalitis virus may cause severe symptoms including abortion and stillbirth in animals such as horses and dogs [93]. Generally, pigs and birds act as reservoirs of the Japanese encephalitis virus, whereas humans and other animals are considered dead-end hosts [93,94]. The enzootic transmission cycle of Japanese encephalitis involves mosquitoes, birds, and pigs. However, there is no evidence that mosquitoes play a significant role in the epidemiology of the encephalitis virus [94].

In the 1930s, the first isolate of the Japanese encephalitis virus was identified from *Culex tritaeniorhynchus* and 30 other mosquito species were later detected [95]. However, mosquitoes from *Culex tritaeniorhynchus*, *Culex gelidus*, *Culex fuscocephala*, *Culex vishnui*, and *Culex quinquefasciatus* (Figure 4c) are well-recognised vectors and capable of transmitting flavivirus both naturally and experimentally [96,97], while mosquitoes from *Aedes*, *Anopheles*, *Armigeres*, and *Mansonia* genera are proven to transmit it experimentally [95,98,99]. Favourable breeding areas for these mosquito species are found to be rice fields, wastewater, and irrigation systems [94,100,101]. Encephalitis is, therefore, most common in countryside areas.

In Vietnam, the Japanese encephalitis virus circulated in both rural and urban areas. In 1951, the first case of Japanese encephalitis virus was isolated in Vietnam [102]. Until 2003, the encephalitis virus was endemic throughout Vietnam, with 1000–3000 cases annually [94,103]. Later, the incidence rate of the encephalitis virus in the Vietnamese population reduced dramatically after the implementation of immunisation and Japanese encephalitis vaccination programmes [102]. In Vietnam, the population of *Culex* species was predominant and its abundance was significantly connected with the density of cattle [104]. *Cx. tritaeniorhynchus*, *Cx. vishnui* s.l., and *Cx. quinquefasciatus* are the major vectors that cause encephalitis in Vietnam [105,106]. Species in the *Cx. vishnui* subgroup, including *Cx. tritaeniorhynchus*, *Cx. vishnui*, and *Cx. pseudovishnui*, breed locally on a large scale in aquatic habitats, i.e., rice areas, puddles, ditches, and cisterns. The most common breeding habitat is the rice-field ecosystem [107]. *Culex quinquefasciatus* is a member of the *Culex pipiens* complex.

### 3.5. Lymphatic Filariasis

Lymphatic filariasis (LF) is a rapidly spreading and neglected tropical disease that causes permanent disruption of the human lymphatic system. It is widely distributed in tropical and subtropical countries in Africa, America, Asia, and the western Pacific [108,109]. Globally, the infection of lymphatic filariasis increased significantly from 120 million cases in 1997 to 56 million cases in 2017 [110]. This disease is caused by mosquito-borne parasitic filarial worms, namely, *Brugia malayi*, *B. timori*, and *Wuchereria bancrofti*, which are transmitted by some mosquito species. About 90% of lymphatic filariasis cases are caused by *W. bancrofti* and other cases by *Brugia* species [111]. The WHO recommended antihelmintic treatment to eliminate lymphatic filariasis.

*Aedes*, *Culex* (predominantly *Cx. quinquefasciatus*) and *Anopheles* species are the principal vectors of lymphatic filariasis causing infections with *W. bancrofti* [112]. *Mansonia* species transmit the disease through *Brugia malayi* and *B. timori*. In 1997, the WHO declared lymphatic filariasis as the second leading disease that causes long-term or permanent disability globally. Integrated vector management programmes have been promoted by the WHO to improve cost-effectiveness, efficacy, ecological soundness, and vector control sustainability [112]. However, the total global burden of this disease continues to grow and its prevalence of endemicity has been confirmed in 76 countries [111].

Lymphatic filariasis (LF) used to be endemic in many parts of Vietnam in the early 1900s, putting millions of people at risk [113,114]. Infections were only caused by *Brugia malayi* and *Wuchereria bancrofti*, in which *B. malayi* made up the majority (over 90%) [113,115]. An estimated 5%–10% of the population became infected in several areas [116]. The prevalence of the disease gradually declined to only 1%–3% in endemic areas [116]. In 2018, Vietnam declared the elimination of LF through the support of the WHO and the United States Agency for International Development (USAID) neglected tropical disease (NTD) programme. To maintain these results, Vietnam continues to conduct operational research to monitor and evaluate, having been recognised by the WHO as having eliminated the disease.

## 4. Other Mosquito-Borne Diseases

### 4.1. Yellow Fever

Yellow fever (Flavivirus) is a common mosquito-borne viral disease distributed mostly in South American and African countries. The clinical features of this disease range from mild febrile illness to lethal symptoms including haemorrhages and liver damage. Annually, 80%–90% of cases are reported from about 44 African countries via epizootic outbreaks [117]. This disease was transmitted to other nations including the United States via slave transport ships in the middle of the 18th century [118]. In 1946, the Pan American Health Organization (PAHO) initiated an eradication campaign to eliminate the vectors causing yellow fever in urban areas of American countries. Unfortunately, this elimination campaign was interrupted and many of the American countries were re-infested by vectors including *Ae. aegypti*, *Ae. africanus*, and Haemagogus janthinomys [118,119]. The main vectors that transmit the yellow fever virus are found to be *Aedes* sp. in Africa, Haemagogus sp. in South America, and *Ae. aegypti* and *Ae. albopictus* in Brazil. In general, Haemagogus leucocelaenus and Sabethes albiprivus are highly susceptible to the yellow fever virus [119].

The disease has not been seen in Vietnam and other Asia counties, except for some suspected cases due to introduction from endemic areas. There have been warnings about the introduction and adaptation to the local *Aedes* mosquito strain of yellow fever virus in some parts of Asia, but this has not been properly verified. Yellow fever belongs to group A in the Law on Prevention and Control of Infectious Diseases in Vietnam [120,121].

### 4.2. Chikungunya Disease

Chikungunya is an alphavirus (Togaviridae) that causes chronic musculoskeletal pain and acute fever in humans. In 1953, chikungunya was first reported in Tanzania and was characterised by severe fever and crippling joint pain [122]. Later, this virus became endemic in Africa through frequent sporadic outbreaks that were reported in the Democratic Republic of Congo, Nigeria, Kenya, Uganda, Senegal, South Africa, and Zimbabwe [123]. In a short time, the disease spread to tropical and temperate zones of Asia through mosquito vectors. In 1958, the first outbreak in Asia was recorded in Bangkok and further spread to Cambodia, Malaysia, Vietnam, and Taiwan in the following years [124]. In recent years, chikungunya cases have been reported in Australia, the Caribbean, South America, Europe, the US, the Middle East, and the Pacific region through travellers from affected countries [125,126]. Chikungunya disease is highly debilitating and its epidemics have a significant economic impact. For instance, within nine months of the first chikungunya infection in the Caribbean island of Saint Martin, it had spread to 22 countries by October 2013 [122].

In southeast Asia, Cambodia has recently appeared in 21 provinces with cases of Chikungunya fever with the rapid spread of the disease on a large scale, the Cambodian government has enhanced the implementation of strict epidemic prevention measures [127]. Vietnam consisted of 10 provinces bordering Cambodia. From 2017 up to now, the southern region has had a total of 22 points of surveillance for Chikungunya virus disease integrated with Dengue hemorrhagic fever surveillance. Currently, there have been no recorded cases of Chikungunya virus infection in some neighbouring localities. The risk of the Chikungunya epidemic in Vietnam is not high [128]. There was no clinical chikungunya case was declared and no CHIKV was found in human febrile patients, the detection of CHIKV in mosquito haplotypes bound to the worldwide movements, to areas with major chikungunya outbreaks, indicates that the threat should be taken seriously and a dedicated surveillance program should be implemented [129]. Dengue and chikungunya, and perhaps other *Aedes*-borne diseases, appear as global threats that should not be addressed at a national or even regional scale but rather at a global scale, with worldwide dimension characterized by permanent exchanges and movements [129].

The extensive spread and frequent epidemics causing severe illness have led to the need for effective drug therapy. Generally, analgesics, anti-inflammatory, antiviral, and antipyretics agents are administered for the treatment of chikungunya fever [130]. In 1984, chloroquine phosphate was observed to be an effective therapeutic agent for chronic chikungunya arthritis [131]. Later, various chikungunya vaccines were developed and used, causing fewer side effects [132,133].

The serological characterisation of blood samples collected in epidemic areas in Tanganyika and Thailand reported *Ae. aegypti* to be the major vector of the chikungunya virus [134,135]. Similarly, experimental studies identified *Ae. albopictus* as the potential vector in Indian Ocean islands [136]. The transmission of the chikungunya virus has differed in Africa and Asia. In Africa, the chikungunya virus is sustained in a sylvatic cycle of *Aedes africanus*, *Ae. aegypti* and *Ae. furcifer-taylori*, while the Asian virus is maintained in a mosquito-human-mosquito cycle of *Mansonia* and *Culex* sp. [136]. However, the *Anopheles* and *Culex* mosquito vectors need further investigation, as they failed to spread the chikungunya virus experimentally [135].

## 5. Mosquito Population Control Strategies

The mosquito control strategy in Vietnam has implemented a number of different methods to control vectors. In the 2000s, measures to improve the environment, clear sewers, clear bushes, etc., were an effective strategy in terms of the economy and savings for the state budget. However, once an epidemic occurs, chemical spraying is applied thoroughly and effectively on a large scale and plays a decisive role. In addition, biological methods have also been applied in Vietnam since the 1990s, although with little effect. They are still applied and include methods such as releasing guppies, carp, and tilapia to eat larvae in ponds, lakes, and ornamental pots [137]. Currently, the use of Wolbachia-carrying mosquitoes appears to be the most effective means of preventing *Aedes* mosquitoes. Trials in Nha Trang in Khanh Hoa showed a significant reduction in the number of mosquitoes in the population, as well as a reduction in the number of dengue cases [138].

### 5.1. Mechanical Methods and Environmental Improvements

Generally, every stagnation group has appropriate areas to breed, mechanical methods eliminate such places in order to minimise the birth of new spores. In addition, by improving the environment the growth of disease-transmitting insects is interrupted and limited [139]. Environmental improvement aims to be detrimental to the disease-causing species, and the ecological imbalance, and maintains this imbalance for as long as possible. For instance, source reduction: limited ponds, cleaning waste, opening sewers, and covering the types of water storage containers (buckets, tanks, drums, and jars), to limit the growth and breeding sites of mosquitoes [140,141]. Overall, mechanical methods and environmental improvement not only kill the vectors but also prevent them from coming into contact with animals and humans. These methods promote high awareness, are inexpensive, and very effective, with the biggest advantages of not causing ecological contamination, and leading to sustainable and proactive effects [142].

### 5.2. Chemical Methods

Over the last few years, chemical agents such as organochloride, pyrethroids, organophosphorus, thiacloprid, imidacloprid, *N*,*N*-diethyl-meta-toluamide, thiamethoxam, *p*-menthane-3,8-diol, fendona (α-cypermethrin), and synthetic chrysanthemum drugs have been given top priority for the inhibition of disease-transmitting vectors [142]. These methods not only limit the vector population but also are remarkably effective at controlling outbreaks in Vietnam. For example, indoor residual spray (IRS) and insecticide-treated nets (ITNs) are currently preferred approaches to malaria vector control [11]. While outdoor and indoor spatial spraying (fogging) is to be used for dengue, zika, and JE vector control [12,143]. The main advantages of these chemical methods are that they are quick-acting and are highly effective, even in large areas. However, this method is limited by the chemical resistance of vectors and environmental pollution [144].

### 5.3. Biological Methods

In this method, either natural enemies of insects are used to destroy vectors (prey) or disease-causing organisms are used to infect with mosquito diseases [145]. For instance, the use of fish to kill larvae, and the release of *Toxorhynchites* mosquito larvae and/or microbes (such as *Bacillus thuringiensis*, *MESOCYCLOPS*) to eat the larvae of *Aedes*, *Anopheles*, *Ceolomomyces*, *Culex*, *Culicinomyces*, *Entomophthora*, *Lagenidium* and *Tolypocladium* [137,138,146,147,148]. Truong et al., 2011, conducted a project about the assessment of the replacement potential of *Ae. aegypti* carrying the biological agent *Wolbachia* starting from April 2013, about 100 mosquitoes carrying *Wolbachia* were released every week in each plot for about 12–18 weeks to replace the population of mosquito transmitted natural disease. Although it is still in the testing process, the results show that this is a safety measure, mosquitoes carrying *Wolbachia* bacteria do not cause any problems for human health or the ecological environment and effectively control the spread of *Wolbachia* bacteria [149]. Biological methods have the advantage of not polluting the environment and are non-toxic to animals and humans. On the other hand, this method is less effective [142].

### 5.4. Genetic Methods

Currently, genetically modified mosquitoes are using as a tool to kill the mosquitoes that transmit dengue fever in Malaysia, Brazil, and Vietnam [150]. In this method, male mosquitoes are genetically engineered and released into the wild to mate with female mosquitoes. The larvae produced after mating with these modified mosquitoes will eventually moult or die at the four instar [151]. This method has been shown to be effective in controlling the transmission of dengue fever. However, genetic methods remain controversial with regards to the emergence of novel species [150]. In case, the Sterile Insect Technique (SIT), the first step of SIT method is to create a generation of sterile male mosquitoes by irradiation and then release them into the wild to mate with female mosquitoes [152]. When released into a population, sterile male mosquitoes mate with female mosquitoes; These female mosquitoes will lay eggs that cannot hatch or develop into larvae, pupa, and adult mosquitoes (cancerous eggs). Since female mosquitoes only mate once during their lifetime, mating with sterile male mosquitoes will make it impossible for them to produce the next generations and the population will dwindle. Commenting on this new method, experts said that the SIT method proved to be highly effective in a large area, limiting pollution compared to having to spray indoors and treat water sources to kill larvae with insecticide [152,153,154,155,156,157].

## 6. An Emerging Method for the Rapid and Reliable Identification of Mosquito Species

Mosquito-borne diseases are a major public health concern worldwide. For effective mosquito control, their accurate identification is a crucial step in differentiating mosquito vectors from non-vectors. Morphological identification is the “gold standard” method, which is widely used for discriminating between mosquitoes using their external characteristics. However, there are several drawbacks to this traditional method, such as the need for standard taxonomic keys, expert entomology knowledge, and the ability to classify species belonging to sibling and cryptic groups [158,159]. Interestingly, most of the significant malaria vectors in the SEA region are members of complexes in which the species are similar or indistinguishable using external morphology. Molecular biology methods such as polymerase chain reaction (PCR), enzyme electrophoresis, and DNA sequencing, have been used to distinguish between homogeneous species [160,161]. Nevertheless, these techniques are also limited due to being time-consuming, exorbitantly priced, and requiring primer-specific targeting for certain species [162,163,164].

Matrix-assisted laser desorption/ionization time-of-flight mass spectrometry (MALDI-TOF MS) has revolutionised the field of clinical microbiology and mycology [162]. Recently, MALDI-TOF MS has been proposed as an alternative and innovative method of overcoming the drawbacks of the above two methods in medical entomology to identify arthropods [164]. This approach has routinely been applied in our laboratory for the rapid and accurate identification of arthropods, including mosquitoes at almost all stages such as eggs, larvae, and adults [165,166,167]. MALDI-TOF MS is an easy-to-use, time-saving, and affordable method compared to earlier methods, and a device may be purchased for medical microbiology and then used in entomology. Its operation is based on the identification of acidic extraction of proteins from an organ of interest. The mass/charge ratio of each protein is measured as it passes across an electric field after the propulsion of the protein molecules from ultraviolet laser desorption. The mass/charge ratio of the proteins generated is a unique protein mass spectrum of the specific sample known as the “protein signature” for reliable species identification. This spectrum is then compared to a database of reference profiles containing the spectra of species that have been officially validated using morphology and molecular techniques.

In Vietnam, MALDI-TOF MS was firstly applied to identify mosquitoes collected in eight areas of the Vietnam Central Highlands, including 22 mosquito species identified from extracted proteins from the legs. In particular, MALDI-TOF MS is able to distinguish between closely related mosquito species, which are impossible to classify using morphology, as was the case of the *Anopheles minimus* [168,169] and *Culex pipiens* complexes [166], which have been reported as being challenging to distinguish from others using morphology and molecular techniques [165,170,171,172]. This innovative technique is capable of successfully identifying Vietnamese mosquitoes at the species level, as well as within the sibling and complex groups. It indicates that MALDI-TOF MS could be used to prevent mosquito misidentification by morphology. 

## 7. Conclusion and Perspectives

Recently, the collection of mosquitoes in Vietnam has mainly focussed on vector species related to the transmission of infectious diseases that cause global health problems. Eighty-nine mosquito species were recognised prior to 1970 and 192 species have been described from 1970 to the present [38]. However, the checklist of Vietnamese mosquitoes is far from being complete as long as further investigations on mosquitoes are carried out. Molecular biology tools have proven to greatly enhance the accurate identification of the microorganisms known to be associated with mosquitoes such as bacteria, protozoa, and arbovirus [173,174,175]. These methods not only confirm the morphological identification but also identify species and contribute to the detection of mosquito-borne microorganisms.

A large number of vectors of the genera *Aedes*, *Anopheles*, and *Culex* have been identified as mosquitoes that transmit diseases caused by parasites, bacteria, and viruses. Interestingly, outbreaks of dengue, Zika, and Japanese encephalitis have occurred in Vietnam and around the world where are *Aedes* spp. mosquitoes are prevalent. Following these viruses, we can hypothesise that a further pathogen transmitted by *Anopheles* and *Aedes* spp. mosquitoes probably appear in Vietnam as *Rickettsia felis* [176]. In fact, an increasing number of reports have implicated *R. felis*, an obligate intracellular bacterial pathogen that is able to cause disease in humans [176,177,178]. The first report of *R. felis* as a human pathogen detected it in two acute undifferentiated fever (AUF) patients and one afebrile adult patient in central Vietnam [179]. The cat flea *Ctenocephalides felis* was first considered to be the only confirmed vector of *R. felis* [176]. Nevertheless, we recently discovered that *Anopheles gambiae*, a major malaria vector in sub-Saharan Africa, was a potential vector for *R. felis* [178]. Additionally, *R. felis* has been detected in *Anopheles sinensis*, *Aedes albopictus*, and *Culex pipiens pallens* mosquitoes from China, where these vectors bite human beings [180]. Therefore, further studies are needed to determine the range of mosquitoes hosting *R. felis* in Vietnam.

## Figures and Tables

**Figure 1 insects-13-01076-f001:**
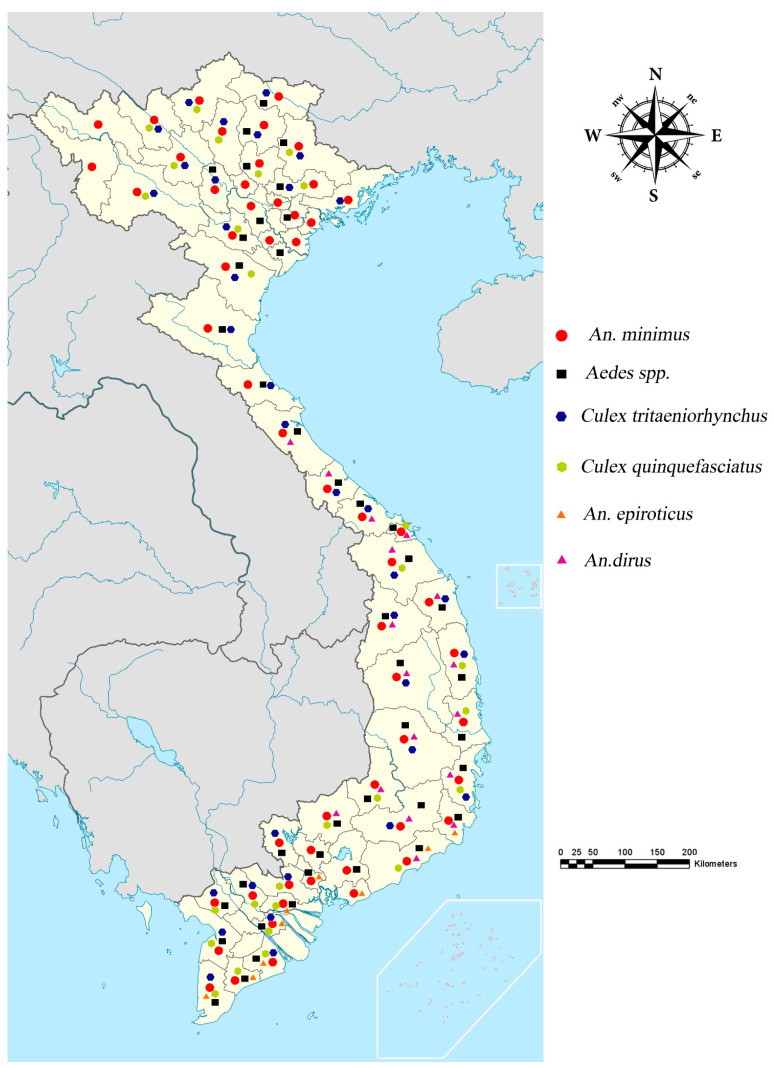
Distribution map of the major vectors.

**Figure 2 insects-13-01076-f002:**
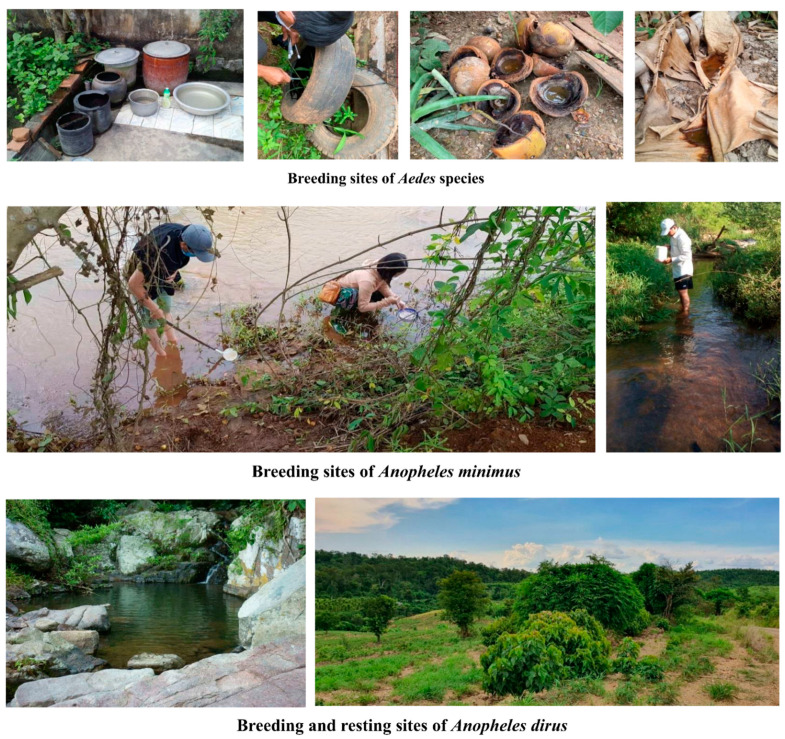
Landscapes of larval breeding and adult resting habitats of *Aedes aegypti*, *Aedes albopictus*, *Culex quinquefasciatus*.

**Figure 3 insects-13-01076-f003:**
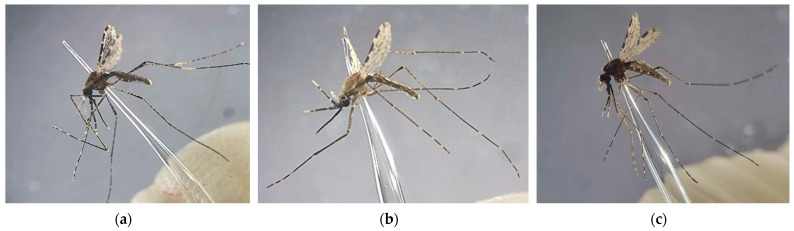
Photographs of three mosquito species which are known to be major malaria vectors in Vietnam: (**a**) *Anopheles dirus*; (**b**) *Anopheles minimus*; (**c**) *Anopheles epiroticus*.

**Figure 4 insects-13-01076-f004:**
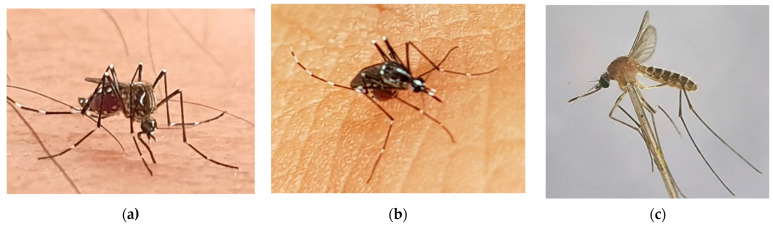
Photographs of three mosquito species which are potential vectors of several viruses, including dengue, Zika, and Japanese encephalitis in Vietnam: (**a**) *Aedes aegypti*; (**b**) *Aedes albopictus*; (**c**) *Culex quinquefasciatus*.

**Table 1 insects-13-01076-t001:** Checklist of *Anopheles*, *Aedes*, and *Culex* species reported in Vietnam.

Genus	Species				
	Before 1970 [34,35,36]		1970-Present [30,31,32,33,37]		
*Anopheles*	*An. aconitus*	*An. philippinensis*	*An. aberrans*	*An. harrisoni*	*An. paraliae*
	*An. annularis*	*An. sinensis*	*An. alongensis*	*An. indefinitus*	*An. pseudojamesi*
	*An. argyropus*	*An. splendidus*	*An. asiaticus*	*An. insulaeflorum*	*An. pseudowillmori*
	*An. barbirostris*	*An. subpictus*	*An. baezai*	*An. interruptus*	*An. pursati*
	*An. campestris*	*An. tessellatus*	*An. baileyi*	*An. jamesii*	*An. rampae*
	*An. crawfordi*	*An.umbrosus*	*An. barbumbrosus*	*An. kochi*	*An. sawadwongporni*
	*An. introlatus*	*An. vagus*	*An. bengalensis*	*An. letifer*	*An. separatus*
	*An. jeyporiensis*	*An. varuna*	*An. cucphuongensi*	*An. lindesayi*	*An. sintonoides*
	*An. karwari*		*An. culicifacies*	*An. monstrosus*	*An. stephensi*
	*An. lesteri*		*An. dangi*	*An. nimpe*	*An. takasagoensis*
	*An. maculatus*		*An. dirus* *	*An. nitidus*	*An. vietnamensis*
	*An. minimus* *		*An. donaldi*	*An. nivipes*	*An. whartoni*
	*An. nigerrimus*		*An. dravidicus*	*An. notanandai*	*An. willmori*
	*An. pallidus*		*An. epiropticus* *	*An. palmatus*	
	*An. peditaeniatus*		*An. gigas*	*An. pampanai*	
*Aedes*	*Ae. eegypti* *	*Ae. mediolineatus*	*Ae. alongi*	*Ae. helenae*	*Ae. novonivveus*
	*Ae. albolineatus*	*Ae. nivens*	*Ae. annandalei*	*Ae. hirsutipleura*	*Ae. pampagensis*
	*Ae. albopictus* *	*Ae. niveoscutellum*	*Ae. andamanensis*	*Ae. gardnerii imitator*	*Ae. patriciae*
	*Ae. alboscutellatus*	*Ae. ostentatio*	*Ae. agrestis*	*Ae. ibis*	*Ae. poecilus*
	*Ae. amesii*	*Ae. pseudalbopictus*	*Ae. caecus*	*Ae. jamesii*	*Ae. prominens*
	*Ae. assamensis*	*Ae. taeniorhynchoites*	*Ae. cancricomes*	*Ae. macfarlanei*	*Ae. saxicola*
	*Ae. aureostriatus*	*Ae. tonkinensils*	*Ae. chrysolineatus*	*Ae. malayensis*	*Ae. thailandensis*
	*Ae. dux*	*Ae. vallistris*	*Ae. culicinus*	*Ae. manhi*	*Ae. uniformis*
	*Ae. imprimens*	*Ae. vexans*	*Ae. desmotes*	*Ae. mediopunctatus*	*Ae. vittatus*
	*Ae. laniger*		*Ae. eldridgei*	*Ae. niveoides*	*Ae. w-albus*
	*Ae. lineatopennis*		*Ae. elsiae*		
*Culex*	*Cx. annulus*	*Cx. peytoni*	*Cx. alienus*	*Cx. malayi*	
	*Cx. bitaeniorhynchus*	*Cx. pseudosinensis*	*Cx. alis*	*Cx. mimulus*	
	*Cx. brevipalpis*	*Cx. pseudovishnui*	*Cx. bernardi*	*Cx. minutissimus*	
	*Cx. fuscanus*	*Cx. quadripalpis*	*Cx. bicornutus*	*Cx. murrelli*	
	*Cx. fuscocephala*	*Cx. quinquefasciatus* *	*Cx. cinctellus*	*Cx. pallidothorax*	
	*Cx. fuscocephalus*	*Cx. raptor*	*Cx. curtipalpis*	*Cx. scanloni*	
	*Cx. gelidus*	*Cx. reidi*	*Cx. edwardsi*	*Cx. sumatranus*	
	*Cx. incomptus*	*Cx. rubithoracis*	*Cx. foliatus*	*Cx. variatus*	
	*Cx. khazani*	*Cx. sinensis*	*Cx. fragilis*	*Cx. viridiventer*	
	*Cx. mimeticus*	*Cx. sitiens*	*Cx. hutchinsoni*	*Cx. vishnui*	
	*Cx. minor*	*Cx. tritaeniorhynchus* *	*Cx. infantulus*	*Cx. wilfredi*	
	*Cx. nigropunctatus*	*Cx. whitei*	*Cx. infula*		
	*Cx. pholster*	*Cx. whitmorei*	*Cx. macdonaldi*		

* Species as a main vector.

**Table 2 insects-13-01076-t002:** Checklist of mosquito species reported in Vietnam, excluding the genera of *Anopheles*, *Aedes*, and *Culex*.

Genus	Species			
	Before 1970 [31,35]	1970-Present [30,31,32,33]		
*Aedeomyia*	*Ad. Catasticta*	-		
*Armigeres*	*Arm. flavus*	*Arm. annulitarsis*	*Arm. durhami*	*Arm. moultoni*
	*Arm. subalbatus*	*Arm. aureolineatus*	*Arm. kuchingensis*	*Arm. pectinatus*
		*Arm. cingulatus*	*Arm. longipalpis*	
		*Arm. dolichocephalus*	*Arm. magnus*	
*Coquillettidia*	*-*	*Cq. crassipes*	*Cq. ochracea*	*Cq. nigrosignata*
*Ficalbia*	*Fi. minima*	*-*		
*Heizmannia*	*-*	*Hz. communis*	*Hz. greenii*	*Hz. scintillans*
		*Hz. complex*	*Hz. persimilis*	
		*Hz. covelli*	*Hz. reidi*	
*Hodgesia*	*Ho. malayi*	*-*		
*Kimia*	*-*	*Km. decorabilis*		
*Lutzia*	*-Lt. raptor*	*Lt. fuscanus*	*Lt. halifaxii*	*Lt. vorax*
*Malaya*	*Ml. jacobsoni*	*Ml. genurostris*	*Malaya* sp.	
*Mansonia*	*Ma. annulifera*	*Ma. annulata*		
	*Ma. crassipes*	*Ma. bonneae*		
	*Ma. nigrosignata*	*Ma. dives*		
	*Ma. ochracea*	*Ma. indiana*		
	*Ma. uniformis* *			
*Mimomyia*	*-*	*Mi. chamberlaini metallica*	*Mi. hybrida*	*Mi. luzonensis*
*Ochlerotatus*		*Och. assamensis*	*Och. macfarlanei*	*Och. saxicola*
		*Och. aureostriatus*	*Och. mikrokopion*	*Och. scatophgoides*
		*Och. chrysolineatus*	*Och. niveoides*	*Och. tonkinensis*
		*Och. dissimilis*	*Och. niveus*	*Och. vigilax*
		*Och. elsiae*	*Och. novoniveus*	*Ochlerotatus* sp.1
		*Och. khazani*	*Och. poicilius*	*Ochlerotatus* sp.2
		*Och. laniger*	*Och. prominens*	*Ochlerotatus* sp.3
		*Och. longirostris*	*Och. pseudotaeniatus*	*Ochlerotatus* sp.4
*Orthopodomyia*	*-*	*Or. albipes*	*Or. andamanensis*	*Or. anopheloides*
*Topomyia*	*-*	*To. Gracilis*	*Topomyia* sp.1	*Topomyia* sp.2
*Toxorhynchites*	*Tx. splendens*	*Tx. albipes*	*Tx. Kempi*	*Toxorhynchites* sp.
*Tripteroides*	*Tp. aranoides*	*Tp. powelli*	*Tp. similis*	*Tp. tenax*
		*Tp. proximus*	*Tp. tarsalis*	
*Udaya*	*-*	*Ud. argyrurus*		
*Uranotaenia*	*Ura. annandalei*	*Ura. bicolor*	*Ura. koli*	*Ura. spiculosa*
	*Ura. campestris*	*Ura. bimaculata*	*Ura. lateralis*	*Uranotaenia* sp.
	*Ura. macfarlanei*	*Ura. bimaculiala*	*Ura. longirostris*	
	*Ura. maxima*	*Ura. demeilloni*	*Ura. lutescens*	
	*Ura. obscura*	*Ura. edwardsi*	*Ura. nivipleura*	
	*Ura. recondita*	*Ura. hongayi*	*Ura. rampae*	
*Verrallina*	*-*	*Ver. andamanensis*	*Ver. consonensis*	*Ver. unca*
		*Ver. butleri*	*Ver. dux*	*Ver. vallistris*
		*Ver. clavata*	*Ver. nigrotarsis*	

## Data Availability

Not applicable.

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
