# Peer review of "Mosquitoes and Mosquito-Borne Diseases in Vietnam"

_insects, 2022, doi:10.3390/insects13121076_

Round 1

Reviewer 1 Report

Dear Authors,

The review is appropriate and provides important information on Vietnam's mosquitoes fauna. I indicate only few suggestions:

line 51-54: it is advisable to increase the bibliogaphy relating to pathogens trasmitted by mosquitoes (i.e. Rift Valley Fever is nort indicate) 

In paragraph 5:  in mechanical and chemical method indicate in more detail which methods used or can be used for the control of mosquitoes. In biological methods enter more information on the use of Wolbachia and fungi as a control method. In genetic methods also include the technique of the SIT (steril Insects Technique). You can seen article of Dahmana and Madiannikov, 2020,  https://doi.org/10.3390/pathogens9040310

Author Response

Authors’ response-Manuscript ID: insects-1978886
Title: Mosquitoes and mosquito-borne diseases in Vietnam
Corresponding author: Xuan Quang Nguyen
Journal: Insects

Reviewer1

Comments and Suggestions for Authors

Dear Authors,

The review is appropriate and provides important information on Vietnam's mosquitoes fauna. I indicate only few suggestions:

line 51-54: it is advisable to increase the bibliogaphy relating to pathogens trasmitted by mosquitoes (i.e. Rift Valley Fever is nort indicate) 

Authors’ response: Dear reviewer, as requested we added “Rift Valley Fever” in the introduction section.

In paragraph 5:  in mechanical and chemical method indicate in more detail which methods used or can be used for the control of mosquitoes.

Authors’ response: as requested in the mechanical method, we modified this sentence as follows “For instance, source reduction: limited ponds, cleaning waste, opening sewers, and covering the types of water storage containers (buckets, tanks, drums, and jars), to limit the growth and breeding sites of mosquitoes”.

In chemical methods, we added to the manuscript following “ For example, indoor residual spray (IRS) and insecticide-treated nets (ITNs) are currently preferred approaches of malaria vector control. While outdoor and indoor spatial spraying (fogging) is to be used for dengue, zika, and JE vector control”.

In biological methods enter more information on the use of Wolbachia and fungi as a control method.

Authors’ response: We added this paragraph to the biological method section following “Truong et al. 2011, conducted a project about the project assessment of the replacement potential of Ae. aegypti carrying the biological agent Wolbachia starting from April 2013, about 100 mosquitoes carrying Wolbachia were released every week in each plot for about 12-18 weeks to replace the population of mosquitoes transmitted natural disease. Although it is still in the testing process, the results show that this is a safety measure, mosquitoes carrying Wolbachia bacteria do not cause any problems for human health or the ecological environment and effectively control the spread of Wolbachia bacteria”.

Using fungi as a control method to reduce mosquito populations. However, there have been no reports of this method in Vietnam. So we just listed the methods that have been applied in the country to fight vectors.

In genetic methods also include the technique of the SIT (steril Insects Technique). You can seen article of Dahmana and Madiannikov, 2020,  https://doi.org/10.3390/pathogens9040310

Authors’ response: In the genetic method section, We added the method of  Sterile Insect Technique to the manuscript following « In case, Sterile Insect Technique (SIT), the first step of the SIT method is to create a generation of sterile male mosquitoes by irradiation, and then release them into the wild to mate with female mosquitoes [153]. When released into a population, sterile male mosquitoes mate with female mosquitoes; These female mosquitoes will lay eggs that cannot hatch or develop into larvae, pupa, and adult mosquitoes (cancerous eggs). Since female mosquitoes only mate once during their lifetime, mating with sterile male mosquitoes will make it impossible for them to produce the next generations and the population will dwindle. Commenting on this new method, experts said that the SIT method proved to be highly effective in a large area, limiting pollution compared to having to spray indoors and treat water sources to kill larvae with insecticide [153-158]’.

Reviewer 2 Report

Dear Authors & Editors,

as far as I can tell, the manuscript entitled "Mosquitoes and mosquito-borne diseases in Vietnam" is the first to provide a complete list of the mosquitoes of Vietnam. That alone would be a valuable contribution and certainly worth publishing. The authors also give an overview of situation of mosquito-borne diseases in Vietnam. This side of the manuscript is quite detailed in some regards, but completely omits some important aspects like monitoring, surveillance and modelling. Below I make some suggestions that may help to alleviate that. However, I am not quite sure whether the authors intend this manuscript to be 1) primarily about mosquitoes with just a bit of additional information about diseases or 2) equally about mosquitoes and diseases. My suggestions were written with option 2) in mind. For option 1) you might want to ignore some of the disease-related suggestions and instead shorten the sections about individual diseases down to what is essential for the situation in Vietnam.

** general points **

What separates the diseases listed in section "4. Other mosquito-borne diseases" to those listed in section "3. Mosquito-associated microorganims and human diseases"? It seems like malaria, dengue, etc. do commonly occur in Vietnam, while yellow fever and chikungunya do not? Then that should be reflected in the section headings.

Before talking about control measures, I am missing a section about surveillance and monitoring strategies for both mosquitoes and diseases. Some ideas that may be worth considering:
- For mosquitoes: are there any programmes to continually monitor mosquitoes, e.g. through trapping? If not, why? Are there any efforts to build a spatial inventory of which species occur where (beyond province level)? Could species distribution models help to identify areas where new species could be found? For example, Outammassine et al. (2021, https://doi.org/10.1111/tbed.14404) suggests that there may be a chance for Ae. japonicus to appear in the North of Vietnam. Similar models exist for many other vector species. The contents of section 6 would fit in here too.
- For diseases/pathogens: How does the monitoring/surveillance system for malaria, dengue etc. work? I'd assume that detected cases are being reported to a central authority in some way? Are/could sentinal animals (typically horses, chicken) being used to detect mosquito-borne zoonoses like JEV, WNV, USUV, ...? What would happen, if a traveller came back from overseas with a CHIKV infection? Would it be detected or mistaken for DENV?
- I'm also missing either a subsection about disease modelling or mentioning of existing models in the individual disease subsections. On the disease side, for dengue alone, a quick web search immediately finds three different forecasting models specifically made for Vietnam (https://www.undp.org/vietnam/projects/integrated-early-warning-dengue-system-viet-nam, https://doi.org/10.1371/journal.pntd.0010509, https://doi.org/10.1371/journal.pmed.1003542), and yet as far as I can see, there is no mention of models anywhere in the whole manuscript. Are those forecasts not being used? If so: what's wrong with them?

Sections on yellow fever and chikungunya (LL 331 ...): From these sections alone, the current status of YFV and CHIKV in Vietnam is unclear to me. Are they endemic, do they occur sporadically, or not at all? If they are not endemic, why is that the case (given that they share vector species with dengue)? Is there a potential for the situation to get worse, also considering climate change? There are modelling studies available that examine the (present and future) climatic suitability for chikungunya (maybe also YFV) on a global scale - what do they suggest for Vietnam?

Section "5. Mosquito population control strategies" is less readable than the preceding ones and would benefit from additional language editing. I don't understand the term "stagnation group" and mosquitoes do not produce spores. The sentence in LL388-389 has strange grammar. There are several more minor grammatical errors/problems throughout this whole section.

Section "6. An emerging method for the rapid and reliable identification of mosquito species", should probably mention the name of the method in the section title (or merge into a section on monitoring and surveillance as suggested above).

** details **

page 2, LL 47-49: widely distributed everywhere, but concentrated in tropics?

page 2, LL 51-54: mosquitoes do not cause diseases, they transmit them

page 2, L 84: Does "fastest spreading" refer to Vietnam or the whole world? This needs a reference. If that's a claim made in ref 24 (Moncayo et al.), it needs a newer reference, as it was published in 2004 - years before the big CHIKV and ZIKV events of the 2010s. Also: (24) should be [24].

page 2, L 90: The world bank reference is OK for me, but please also add a peer-reviewed scientific reference or a relevant section from the latest IPCC reports.

page 7, LL 194-196: What does "most prominent" mean here? most cases? most deaths? most media coverage? Also, does "most prominent" refer to Vietnam, the whole world, or the tropical zone? Needs a reference, too.

Table 1: The title/caption "Checklist of mosquito species reported in Vietnam" suggest that this is a complete list, but in only contains species of Culex, Anopheles and Aedes. Change that to something like "Checklist of Anopheles, Aedes, and Culex species reported in Vietnam".

Table 2: Captions of tables (and figures) should contain the necessary information to understand the contents of the table without looking at the main text or other tables. When viewed in isolation, the title/caption "Checklist of other mosquito species reported in Vietnam" is confusing. Change to something like "Checklist of mosquito species reported in Vietnam, excluding the genera of Anopheles, Aedes, and Culex".

Table 1 and 2: maybe consider marking competent vector species somehow, e.g. with an asterisk (*)?

All figures are of very low resolution - please check before publishing.

Author Response

Authors’ response-Manuscript ID: insects-1978886
Title: Mosquitoes and mosquito-borne diseases in Vietnam
Corresponding author: Xuan Quang Nguyen
Journal: Insects

Reviewer2

Comments and Suggestions for Authors

Dear Authors & Editors,

as far as I can tell, the manuscript entitled "Mosquitoes and mosquito-borne diseases in Vietnam" is the first to provide a complete list of the mosquitoes of Vietnam. That alone would be a valuable contribution and certainly worth publishing. The authors also give an overview of situation of mosquito-borne diseases in Vietnam. This side of the manuscript is quite detailed in some regards, but completely omits some important aspects like monitoring, surveillance and modelling. Below I make some suggestions that may help to alleviate that. However, I am not quite sure whether the authors intend this manuscript to be 1) primarily about mosquitoes with just a bit of additional information about diseases or 2) equally about mosquitoes and diseases. My suggestions were written with option 2) in mind. For option 1) you might want to ignore some of the disease-related suggestions and instead shorten the sections about individual diseases down to what is essential for the situation in Vietnam.

Authors’ response: Dear Reviewer, we agree with the option 2: equally about mosquitoes and diseases

** general points **

What separates the diseases listed in section "4. Other mosquito-borne diseases" to those listed in section "3. Mosquito-associated microorganims and human diseases"? It seems like malaria, dengue, etc. do commonly occur in Vietnam, while yellow fever and chikungunya do not? Then that should be reflected in the section headings.

Authors’ response: Exactly, in the Mosquito-associated microorganism and human diseases section, we listed the mosquito-borne diseases that commonly occur in Vietnam. For example, the outbreak of dengue fever was currently very serious with more than 250,000 cases and over 100 cases of death (10/2022). Today, dengue is endemic throughout the southern region and central coast, and in densely populated hubs such as Ho Chi Minh City and Hanoi. For malaria, although there are encouraging reports of declined malaria morbidity and mortality in Vietnam in the last two decades. However, malaria prevalence in central and southern provinces is still high. From April to September 2019 there was an 18% increase in confirmed cases nationwide and an 39% increase in Plasmodium falciparum compared to the same period in 2018 (4813 cases).

In the other mosquito-borne diseases section, we listed the mosquito-borne diseases that are commonly less and/or currently have been no reported cases in Vietnam

Before talking about control measures, I am missing a section about surveillance and monitoring strategies for both mosquitoes and diseases. Some ideas that may be worth considering:
- For mosquitoes: are there any programmes to continually monitor mosquitoes, e.g. through trapping? If not, why?

Authors’ response: Entomological monitoring and surveillance is carried out as a routine method in our vector control strategies by collecting indoor and outdoor resting mosquitoes using different catch methods. Then, a bioassay method or susceptibility test is used to determine the insecticide resistance status of the vector. Mosquitoes are identified and processed for parasite identification, vector incrimination, and sibling species determination. So, understanding vector density, distribution, insecticide resistance, vector incrimination, infection status, and identification of sibling species are all important components of vector control measures.

In this review paper, we would like to focus on providing a complete checklist of Vietnamese mosquitoes along with an overview of mosquito-borne diseases in Vietnam, and some preventive measures for mosquitoes in Vietnam. For this reason, we are interested in your suggestion and will consider including it in a future study focusing on vector control and investigation measures in Vietnam.

Annually, we have been receiving several support progammes such as from the Government of Vietnam, the Global Fund to Fight Malaria (regarding Vietnam’s Regional Artemisinin-resistance Initiative (RAI3) grant) to monitor mosquitoes

Mosquito collections were conducted every month, based on adult trapping through dry ice-baited CDC traps, CDC light traps, BG-Sentinel traps, and animal bait nep traps, while larval sampling through dippers and nets, and ovitrapping.

Are there any efforts to build a spatial inventory of which species occur where (beyond province level)? Could species distribution models help to identify areas where new species could be found? For example, Outammassine et al. (2021, https://doi.org/10.1111/tbed.14404) suggests that there may be a chance for Ae. japonicus to appear in the North of Vietnam. Similar models exist for many other vector species. The contents of section 6 would fit in here too.

Vietnam is a tropical country that is suitable for the development of arthropods, including mosquitoes. So, understanding the spatial extent of potential vector species, as well as their abundance and seasonal activity, is critical for estimating MBD risk levels and enabling better surveillance and controversy targeting. For example, since 2008, Vietnam started an inventory of 15 provinces that have been reported to be indigenous mosquito epidemic areas in order to gather basic information on the composition, geographical distribution, and biodiversity preferences of mosquito species. The main malaria vector Anopheles minimus was one of the most commonly collected mosquito species. The abundance of An. minimus with biodiversity features and to provide distribution maps in Vietnam. Mosquito data were collected using a cross-sectional study design in Vietnam, from October to November 2003–2004, and based on previous records. The latitude/longitude position and altitude of the sites were measured with a Global Positioning System (GARMIN Etrex, Hampshire, UK). Mosquito densities were incorporated into a database linked to the Geographic Information System package ArcView (ESRI, Redlands, California).

 [The citation of Claire Garros et al. 2008 Distribution of Anopheles in Vietnam, with particular attention to malaria vectors of the Anopheles minimus complex]

- For diseases/pathogens: How does the monitoring/surveillance system for malaria, dengue etc. work? I'd assume that detected cases are being reported to a central authority in some way?

Authors’ response: In Vietnam, the Malaria and Dengue Case Management and Monitoring System is operated at the grassroots level. From the commune level, cases are reported to the district and provincial levels. It was then managed by the National Institutes of Malaria and the Department of Preventive Medicine under the Ministry of Health.

Are/could sentinal animals (typically horses, chicken) being used to detect mosquito-borne zoonoses like JEV, WNV, USUV, ...?

JEV, WNV, USUV has never appeared in Vietnam, however, the source of infection (humans or sick animals) and vectors may enter and spread into our country. Regular epidemiological surveillance measures combined with border health quarantine, including animal quarantine, are important to promptly detect the source of infection from endemic areas of the world. Combined with veterinary and environmental authorities in surveillance and control of diseases in horses, ungulates, and wild birds.

 What would happen, if a traveller came back from overseas with a CHIKV infection? Would it be detected or mistaken for DENV?

When a tourist returns from a country infected with CHIKV, in Vietnam, there is an international control program such as implementing an Internet surveillance network through which to know information about the epidemic during the time that the traveler is in an epidemic area. disease, alerting clinicians to identify CHIKV and allowing them to do more rapid tests to avoid confusion with symptoms of dengue.

At present, there are no statistics on how many cases are detected and how many are misdiagnosed with DENV

- I'm also missing either a subsection about disease modelling or mentioning of existing models in the individual disease subsections. On the disease side, for dengue alone, a quick web search immediately finds three different forecasting models specifically made for Vietnam (https://www.undp.org/vietnam/projects/integrated-early-warning-dengue-system-viet-nam,https://doi.org/10.1371/journal.pntd.0010509, https://doi.org/10.1371/journal.pmed.1003542), and yet as far as I can see, there is no mention of models anywhere in the whole manuscript. Are those forecasts not being used? If so: what's wrong with them?

In this manuscript, we prioritize the mosquito species associated with each mosquito-borne disease. The issue of early warning is currently being concerned by the government and implemented

Sections on yellow fever and chikungunya (LL 331 ...): From these sections alone, the current status of YFV and CHIKV in Vietnam is unclear to me. Are they endemic, do they occur sporadically, or not at all? If they are not endemic, why is that the case (given that they share vector species with dengue)? Is there a potential for the situation to get worse, also considering climate change? There are modelling studies available that examine the (present and future) climatic suitability for chikungunya (maybe also YFV) on a global scale - what do they suggest for Vietnam?

Authors’ response: We added this paragraph in the Yellow fever section as follows “The disease has not been seen in Vietnam and other Asia countries, except for some suspected cases due to introduction from endemic areas. There have been warnings about the introduction and adaptation to the local Aedes mosquito strain of yellow fever virus in some parts of Asia, but this has not been properly verified. Yellow fever belongs to group A in the Law on Prevention and Control of Infectious Diseases in Vietnam".

In the Chikungunya section, we added this context "In southeast Asia, Cambodia has recently appeared in 21 provinces with cases of Chikungunya fever with the rapid spread of the disease on a large scale, the Cambodian government has enhanced the implementation of strict epidemic prevention measures [129]. Vietnam consisted of 10 provinces bordering Cambodia. From 2017 up to now, the southern region has had a total of 22 points of surveillance for Chikungunya virus disease integrated with Dengue hemorrhagic fever surveillance. Currently, there have been no recorded cases of Chikungunya virus infection in some neighboring localities. The risk of the Chikungunya epidemic in Vietnam is not high [130]. There is no clinical chikungunya case was declared and no CHIKV was found in human febrile patients, the detection of CHIKV in mosquito haplotypes bound to the worldwide movements, to areas with major chikungunya outbreaks, indicates that the threat should be taken seriously and a dedicated surveillance program should be implemented [131]. Dengue and chikungunya, and perhaps other Aedes-borne diseases, appear as global threats that should not be addressed at a national or even regional scale but rather at a global scale, with worldwide dimension characterized by permanent exchanges and movements"

Lessons learned for Vietnam to manage chikungunya is Border Health Quarantine in Vietnam. All citizens self-declared illness when transit. In addition, mandatory quarantine and insecticidal measures are applied to ships, aircraft, and road vehicles coming from places where yellow fever is present. Combined animal quarantine for imported primates (monkeys, gibbons, orangutans), monitoring 7-14 days after leaving the yellow fever area. Otherwise, a certificate of vaccination is required. yellow fever strains for people entering from yellow fever endemic areas and Vietnamese people about to go to yellow fever epidemic areas.

Section "5. Mosquito population control strategies" is less readable than the preceding ones and would benefit from additional language editing. I don't understand the term "stagnation group" and mosquitoes do not produce spores. The sentence in LL388-389 has strange grammar.

Authors’ response: It means that some ponds have standing water

There are several more minor grammatical errors/problems throughout this whole section.

The manuscript has been professionally edited by the TRADONLINE company.

Section "6. An emerging method for the rapid and reliable identification of mosquito species", should probably mention the name of the method in the section title (or merge into a section on monitoring and surveillance as suggested above).

Authors’ response: In terms of vector surveillance and monitoring in Vietnam

Every year, by organizing campaigns, the Minister of Health of Vietnam calls on authorities, mass organizations, and people across the country to join hands in the prevention and control of dengue fever through simple actions. Simple and practical to "no larvae/pupa, no dengue".

In addition to implementing synchronous measures in dengue control, such as: deploying a network of collaborators in dengue prevention; handling outbreaks. Actively spraying and stamping out epidemics on a large scale, the "Community-based anti-dengue prevention, and control campaign" is a solution to help mobilize the entire community's resources and is highly proactive. In order to raise people's awareness of actively killing the larvae in households and enhance the role and participation of authorities at all levels, departments, agencies, and mass organizations in the prevention and control of dengue fever. Create a mass, comprehensive movement in the community to kill mosquitoes and mosquito vectors that transmit dengue, reduce disease transmission and reduce the number of cases. Communication programs on raising dengue awareness should be repeated all year round and target particular groups of adolescents, younger adults, landlords, and migrants from other provinces to improve their knowledge and encourage them to implement preventive measures against dengue fever.

As we mentioned above, in this review paper, we prioritize the mosquito species associated with each mosquito-borne disease. The issue of early warning is currently being concerned by the government and implemented.

** details **

page 2, LL 47-49: widely distributed everywhere, but concentrated in tropics?

It means that the mosquito concentrated in the tropical Zone

page 2, LL 51-54: mosquitoes do not cause diseases, they transmit them

Authors’ response: The reviewer is right, we have corrected the paragraph as follows “The mosquito vectors of the Anopheles genus can transmit malaria and filariasis, while the Aedes genus can transmit dengue virus (DENV), chikungunya virus (CHIKV), Zika virus (ZIKV), yellow fever virus (YFV), and the Culex genus can transmit filariasis, Japanese encephalitis virus (JEV), West Nile virus (WNV), and Rift Valley Fever (RVF)” in the introduction section.

page 2, L 84: Does "fastest spreading" refer to Vietnam or the whole world? This needs a reference. If that's a claim made in ref 24 (Moncayo et al.), it needs a newer reference, as it was published in 2004 - years before the big CHIKV and ZIKV events of the 2010

Authors’ response: The “fastest spreading” refer to “in the whole world” to clarify the paragraph in the introduction section.

We changed “the citation of Moncayo et al., 2004” to “the citation of GCRF 2022” in the introduction section.

Also: (24) should be [24].

We also corrected a reference with a round blankets “( )” to a square blankets “[ ]” in the introduction section

page 2, L 90: The world bank reference is OK for me, but please also add a peer-reviewed scientific reference or a relevant section from the latest IPCC reports.

Authors’ response: As requested by the reviewer, we added the citation of the latest IPCC report (The IPCC Sixth Assessment Report on Climate Change Impacts - 2022 - Population and Development Review) in the introduction section

page 7, LL 194-196: What does "most prominent" mean here? most cases? most deaths? most media coverage? Also, does "most prominent" refer to Vietnam, the whole world, or the tropical zone? Needs a reference, too.

The word means popular. Also, refer to the tropical zone

We also added the citation of WHO. Dengue and severe dengue2022” in this paragraph

Table 1: The title/caption "Checklist of mosquito species reported in Vietnam" suggest that this is a complete list, but in only contains species of Culex, Anopheles and Aedes. Change that to something like "Checklist of Anopheles, Aedes, and Culex species reported in Vietnam".

Authors’ response: The title of the table 1 has been modified as follows “Checklist of Anopheles, Aedes, and Culex species reported in Vietnam” in Table 1

Table 2: Captions of tables (and figures) should contain the necessary information to understand the contents of the table without looking at the main text or other tables. When viewed in isolation, the title/caption "Checklist of other mosquito species reported in Vietnam" is confusing. Change to something like "Checklist of mosquito species reported in Vietnam, excluding the genera of Anopheles, Aedes, and Culex".

Authors’ response: As requested by the reviewer, we modified the title of table 2 as follows “Checklist of mosquito species reported in Vietnam, excluding the genera of Anopheles, Aedes, and Culex” in Table 2

Table 1 and 2: maybe consider marking competent vector species somehow, e.g. with an asterisk (*)?

Authors’ response: we added an asterisk in each main vector in Tables 1 and 2.

All figures are of very low resolution - please check before publishing.

Authors’ response: Our figures already put at 300 and 600 dpi

We modified “Fig 1A” to “Fig 1”, “Fig 1B” to “Fig 2”

“Figure 2” refers to “Figure 3” and “figure 3” refers to “figure 4”
